# ExoProK: A Practical Method for the Isolation of Small Extracellular Vesicles from Pleural Effusions

**DOI:** 10.3390/mps4020031

**Published:** 2021-05-11

**Authors:** Dionysios Antonopoulos, Irene Tsilioni, Sophia Tsiara, Eirini Moustaka, Spyridon Ladias, Garyfallia Perlepe, Theoharis C. Theoharides, Konstantinos I. Gourgoulianis, Nikolaos A. A. Balatsos

**Affiliations:** 1Department of Biochemistry and Biotechnology, University of Thessaly, Viopolis, 415 00 Larissa, Greece; diadonop@uth.gr (D.A.); stsiara@bio.uth.gr (S.T.); moustaka.renia@gmail.com (E.M.); 2Department of Immunology, Tufts University School of Medicine, 136 Harrison Avenue, Suite J304, Boston, MA 02111, USA; Eirini.Tsilioni@tufts.edu (I.T.); theoharis.theoharides@tufts.edu (T.C.T.); 3Respiratory Medicine Department, Faculty of Medicine, University of Thessaly, Viopolis, 411 10 Larissa, Greece; spyrosladias@yahoo.gr (S.L.); perlepef19@gmail.com (G.P.)

**Keywords:** extracellular vesicles, exosomes, pleural effusion, proteinase K, RNA biomarkers

## Abstract

Extracellular vesicles (EVs) are cell-secreted, lipid membrane-enclosed nanoparticles without functional nucleus. EV is a general term that includes various subtypes of particles named microvesicles, microparticles, ectosomes or exosomes. EVs transfer RNA, DNA and protein cargo between proximal and distant cells and tissues, thus constituting an organism-wide signal transduction network. Pathological tissues secrete EVs that differ in their cargo composition compared to their healthy counterparts. The detection of biomarkers in EVs from biological fluids may aid the diagnosis of disease and/or monitor its progression in a minimally invasive manner. Among biological fluids, pleural effusions (PEs) are integrated to clinical practice, as they accompany a wide variety of lung disorders. Due to the proximity with the pleura and the lungs, PEs are expected to be especially enriched in EVs that originate from diseased tissues. However, PEs are among the least studied biofluids regarding EV-specialized isolation methods and related biomarkers. Herein, we describe a practical EV isolation method from PEs for the screening of EV RNA biomarkers in clinical routine. It is based on a Proteinase K treatment step to digest contaminants prior to standard polyethylene-glycol precipitation. The efficiency of the method was confirmed by transmission electron microscopy, nanoparticle tracking analysis and Western blot. The reliability and sensitivity of the method towards the detection of EV-enriched RNA biomarkers from multiple PEs was also demonstrated.

## 1. Introduction

Pleural effusion (PE) is the excessive accumulation of fluid in the pleural space. It is distinguished in exudates and transudates; exudates are caused by active inflammation in the pleura, while transudates are caused by an imbalance between vascular hydrostatic and oncotic pressures [1,2]. PEs accompany a wide variety of pleura disorders including malignant conditions such as carcinomas of the lung and the pleural cavity. The biochemical and cellular components of PEs are routinely utilized in standard clinical laboratory practice. Although such clinical tests typically reach 100% specificity, they average only 60% sensitivity for diagnosis of malignancies [3,4]. Therefore, it is necessary to develop additional tests to supplement the established ones. As with all biological fluids, PEs, besides cells and macromolecules, also contain EVs that hold much untapped potential as carriers of biomarkers for lung diseases [5]. EVs are particles naturally released from a cell delimited by a lipid bilayer without a functional nucleus [6]. The term EV is a collective one that includes various subtypes of particles named microvesicles, microparticles, ectosomes, exosomes, etc. [6]. EVs are classified according to their physical characteristics, particularly to their size, as small EVs (<200 nm) and medium/large EVs (>200 nm), although they have also been described based on their biochemical composition (e.g., CD9^+^-EVs, CD63^+^-EVs, etc.) or their cell origin (oncosomes, apoptotic bodies, etc.) [6,7]. EVs are secreted by all cell types and are present in all biological fluids, and it is now widely accepted that EVs are intracellular signaling vehicles [8,9,10]. The donor cells sort specific species of biological molecules (RNA, DNA, lipid, metabolite and protein) inside EVs. Once released in the circulation, they deliver their cargo to non-random tissues via a variety of uptake pathways, based on interactions between membrane receptors, ligands or contact proteins of recipient cells and proteins (including tetraspanins, lectins, proteoglycans and integrins) of the EVs [11]. Then, they release their cargo inside the recipient cells influencing their gene expression and phenotype [12]. Since all cells secrete EVs, they participate in an organism-wide signal transduction network [8,10,12]. Consequently, cells from diseased tissues or organs secrete EVs that differ in their cargo composition compared to the healthy cells that they originate from [13]. The utilization of PE derived EV biomarkers to supplement established examinations presents much more potential compared to other biofluids due to the anatomical proximity of the PE to the cancerous tissue. Cancer cells of the pleural epithelium secrete EVs, that are directly deposited in the resulting pleural effusion. Similarly, the lung is directly connected by the thoracic lymphatic system to the pleural cavity, where cancer-derived EVs can easily traverse to the PE. Additionally: (a) large volumes of PE are typical of malignant conditions; (b) EVs are generally abundant in biological fluids; and (c) amplification based molecular techniques are inherently sensitive.

A key requirement to integrate EV-based liquid biopsy as part of routine clinical examinations is the development of EV isolation methods that are practical to screen routinely for cargo biomarkers in clinical laboratories. Such methods must be reliable, inexpensive, automated and capable of processing large amounts of samples simultaneously, as well as be straightforward enough to be performed by non-specialized personnel [14]. Among the most well-established, inexpensive, efficient and straightforward methodologies to isolate EVs is by precipitation after the addition of volume excluding polymer solutions such as polyethylene glycol (PEG) [15]. Various commercial kits are also based on volume-excluding polymer-mediated precipitation [16] (Table 1). The inherent disadvantage of these isolation techniques is the low purity of the resulting EV pellets and the subsequent co-precipitation of non-EV macromolecules [15]. Moreover, biological fluids such as pleural fluid typically contain an abundance of non-EV nanoparticles such as high- and low-density lipoproteins (HDL and LDL, respectively), further complicating the isolation of EVs with adequate purity [17,18]. These techniques are hampered by the presence of contaminants that limit their reliability for downstream analyses, a dire flaw in clinical diagnostics [14,19].

Two studies have optimized the parameters of sEV isolation by PEG precipitation from cell cultured media [30,31]. The efficiency of sEV isolation increases in proportion to the molecular weight and the final concentration of the PEG solution added, as well as the incubation time after the addition of PEG. The maximum efficiency is achieved by the addition of PEG ranging from 6000 to 8000 kDa in solutions to a final concentration of 10–12% *w*/*v* incubated for 8 h [30]. When compared to ultracentrifugation, PEG precipitation is the more efficient method. However, the purity is significantly inferior, and it has been stressed that further purification steps must be performed if the sEV cargo is to be further analyzed by high throughput techniques [30,31].

In our current work, we report an sEV isolation method from pleural fluid that fulfils all of the abovementioned criteria. The method is based on two previous studies [30,31] and uses conditions to ensure maximum efficiency of EV isolation. We integrated a Proteinase K digestion step in our method to counteract the inherently low purity of PEG precipitation and the abundance of contaminants in the PE. The Proteinase K step and a subsequent centrifugation clearly depleted the sample from contaminating proteins. In addition, non-EV RNA is removed with a subsequent RNase A step. We confirmed the efficiency of our method by: (a) visualizing the sEV under electron microscopy; (b) nanoparticle tracking analysis; (c) detection of TSG101 sEV marker by Western blot; and (d) sEV contained RNAs by qPCR. Finally, we compared the effectiveness of our method with a well-established column-affinity based method for the detection of sEV RNAs [23]. Using ExoProK, we detected sEV-enriched RNAs, namely miR-29a-5p, miR-484 and miR-21-5p, as well as U6 snRNA. We named the method ExoProK to underline the significance of Proteinase K to deplete contaminants from PEs prior to sEV precipitation by PEG.

## 2. Materials and Methods

### 2.1. Patients and Sample Collection

Pleural effusion samples (n = 20, Table 2) were collected from hospitalized patients diagnosed with pleural effusion, due to malignant and non-malignant diseases, irrespective of past, ongoing treatment or any other factor. The samples were collected between January 2017 and November 2018 at the Respiratory Medicine Department of University Hospital of Larissa. The study was approved by the local ethics committee, and all subjects gave their written informed consent. The etiology of pleural effusions was diagnosed with pleural biopsy as well as standard biochemical and cytological examinations with methods we described in our earlier study [2].

Pleural fluid was collected after regulation of anticoagulant drugs, if the patient was receiving any. A 10 mL blood sample was collected and processed for measuring full blood count, prothrombin time (PT), activated partial thromboplastin time (aPTT), International Normalized Ratio (INR), glucose, total proteins, albumin, LDH and CRP. If the measured INR was <2, diagnostic puncture of the pleural cavity was performed. The patient was placed in sited position, the spot of puncture was determined with thoracic ultrasound, the desired area was sterilized with povidone iodine and ethyl alcohol solution alternatively 3 times and then was punctured with a 21G needle. Then, approximately 40 mL of pleural fluid were collected with two 20 mL syringes. The sample was aliquoted for cytological and biochemical examinations and sEV isolation.

### 2.2. sEV Isolation

Freshly isolated pleural effusions (PE) were immediately centrifuged in gel and clot activator tubes for 15 min at 3000 RCF and 4 °C. After this step, the samples can be frozen at −80 °C. The supernatants were further centrifuged for 30 min at 6000 RCF and 4 °C. Proteinase K (11912312; Macherey Nagel, Dueren, Germany) was added to the supernatants at a final concentration of 0.67 μg/mL and incubated in a heat block at 60 °C for 1 h with occasional inversions every 15 min. The mixture was centrifuged for 30 min at 6000 RCF and 4 °C and the supernatants were transferred to microcentrifuge tubes. PMSF was then added to inhibit the activity of proteinase K at a final concentration of 1 mM. Then, 1.5 mL of the sample supernatants were centrifuged for 30 min at 16,000 RCF and 4 °C to remove larger vesicles. A 50% *w*/*v* solution of polyethylene glycol in filter-sterilized PBS of an average molecular weight of 8000 kDA (Merck, Darmstadt, Germany) was added to a final concentration of 12% *w*/*v*, and the solutions were vortexed until they became visibly homogenous. The solutions were then placed in a tube shaker (Thermomixer comfort, Eppendorf, Hamburg, Germany) overnight (approximately 12 h) at 400 RPM and 4 °C. The next day the mix was centrifuged for 1 h at 23,000 RCF at 4 °C to pellet the sEV. The pellets were then solubilized in 240 μL PBS, and RNase A (Sigma cat # R5125) was added to a final concentration of 1 μg/μL and incubated at 37 °C for 30 min. The sEV suspensions can be stored at −80 °C or utilized for further experiments (Figure 1). sEVs were also isolated from 1.5 mL of pre-cleared pleural effusion supernatant by the exoEasy Maxi Kit (Qiagen, Valencia, CA, USA) according to the manufacturer’s instructions. PE samples were mixed with XBP buffer (EMD) and transferred to the spin columns. The EVs adsorbed on the column were washed with buffer XWP (EMD), eluted with 400 μL Buffer XE and filtered through 0.22-micron syringe filters to exclude larger vesicles. The sEV were then ready for further analysis or stored at −80 °C.

### 2.3. Transmission Electron Microscopy

A single drop (5 μL) of isolated EV sample was placed on a copper grid for 1 min. The grid was then washed with a drop of DEPC water, and the excess liquid was removed with a Whatman paper. The grids were stained by incubating with 2 μL of uranyl formate 0.75% for 30 s. After removing the excess uranyl formate in a similar manner, the grids were scanned with a TecnaiG2 Spirit BioTWIN TEM (FEI, Hillsboro, OR, USA), and images were taken with an AMT 2k CCD camera at a magnification of 18,500–30,000×.

### 2.4. Protein Quantification and Western Blot Analysis

The EV suspensions were lysed by the addition of RIPA and the concentration of sEV proteins was determined by the bicinchoninic acid (BCA) assay (Thermo Fisher Inc., Rockford, IL, USA) using bovine serum albumin (BSA) as standard. sEV positive marker TSG101 and sEV negative marker albumin was detected by Western blot. Ten micrograms of proteins derived from lysed sEV were separated on 4–12% NuPAGE Bis-Tris starting at 65 V for 45 min and then increased to 90 V for another 30 min. Proteins were then electrotransferred onto a nitrocellulose membrane (Bio-Rad, Hercules, CA, USA) followed by blocking for 1 h using 5% Non-fat dry milk in Tris-buffered saline containing 0.05% Tween-20. The membrane was then incubated overnight at 4 °C with TSG101 (ab #30871; Abcam, Cambridge, MA, USA) or human serum albumin at 1:1000 dilution (sc-271605; Santa Cruz Biotechnology Inc. Dallas, TX, USA). For detection, the membranes were incubated with an appropriate secondary horseradish peroxidase (HRP)-conjugated antibody (System Biosciences, Palo Alto, CA, USA) at 1:20,000 dilution for 1 h at room temperature, and the blots were visualized by enhanced SuperSignal West Pico Chemiluminescence for TSG101 (ThermoFisher Scientific, Carlsbad, CA, USA) and by the SNOW software in Amersham ImageQuant^TM^ 800 biomolecular imager (Cytiva, Marlborough, MA, USA) for serum albumin.

### 2.5. Nanoparticle Tracking Analysis

A NanoSight LM10 (Malvern Panalytical Ltd., Malvern, UK) was used to measure the size distribution of the isolated sEV preparations; it was performed at the Nanosight Nanoparticle Sizing and Quantification Facility at Massachusetts General Hospital (Boston, MA, USA). Briefly, 1 μL of sEV sample was diluted 1:299 in 1 × PBS, the 405-nm laser was used, and the detection threshold was set to 8. Next, the mixture was placed into the 1 mL injector and injected into the nanoparticle tracking analyzer. The Brownian motion of sEVs was recorded and tracked, generating size distribution data by applying the Stokes–Einstein equation. Three 30 s measurements were performed and averaged from each sample.

### 2.6. RNA Isolation and Precipitation

Total RNA was extracted from sEV suspensions with Trizol LS (ThermoFisher Scientific, Carlsbad, CA, USA) coupled with Monarch total RNA miniprep kit (NEB, Ipswich, MA, USA), the isolated RNA was subsequently precipitated with sodium acetate and ethanol and its concentration measured by a nanodrop 2000c spectrophotometer. Then, 750 μL of Trizol LS were directly added to 250 μL of frozen sEV suspension and mixed until the solution became homogeneous, then incubated at room temperature for 20 min. Next, 0.1 V of 1-bromo-3-chloropropane (Merck, Darmstadt, Germany) was added, and the tubes were continuously inverted by hand for about a minute then left at room temperature for at least 15–30 min until phases separate completely. Tubes were centrifuged at 12,000 RCF for 15 min at 4 °C and the aqueous phase was carefully separated and mixed thoroughly with an equal volume of 100% ethanol then transferred to a Monarch RNA column and on-column DNase digestion was performed according to the manufacturer’s protocol (NEB, Ipswich, MA, USA). RNA was eluted in 100 μL of DEPC water to ensure maximum RNA recovery. To concentrate the eluted RNA by precipitation, 2.5 V of 100% ethanol, 0.1 V of 3 M CH_3_COONa and 5 μL of molecular biology grade glycogen (20 mg/mL) (R0561, ThermoFisher Scientific, Carlsbad, CA, USA) were added in that order and mixed. The solution was incubated at −20 °C for at least 16 h, RNA was precipitated by centrifugation 23,000 RCF/15 min/4 °C, the supernatant was carefully removed and the pellet was washed with 2.5 V of 70% ethanol without resuspending the pellet. After incubating the RNA for 2 min at RT, RNA was precipitated at 23,000 RCF/15 min/4 °C, the supernatant was removed and the RNA pellet was dried in a heatblock placed in a PCR hood that is set to 40 °C [32]. The RNA pellet was resuspended in 7 μL of DEPC water, 1 µL of which was utilized to measure its concentration with a NanoDrop 2000c spectrophotometer (Thermo Scientific, Carlsbad, CA, USA). RNA concentrations typically ranged 100–250 ng/μL.

### 2.7. cDNA Synthesis-qPCR

sEV RNA was 3′ polyadenylated and reverse transcribed into cDNA by the mir-X first strand cDNA synthesis kit (Takara) following the manufacturers protocol; 3.75 μL of RNA were mixed with 5 μL of mRQ buffer and 1.25 μL of mRQ enzyme the enzymatic reactions were carried out at 37 °C for 1 h and then stopped at 85 °C for 5 min the cDNA mix was subsequently diluted to 100 μL with DEPC water. sEV miRNAs was detected by qPCR using SYBR chemistry (SYBR select mix Applied Biosystems). The entire sequence of the mature miRNAs was obtained from www.mirbase.com (access date: 17 June 2019) and was used as miR specific primers. miR-29a-5p, 5′ GGG GACTGATTTCTTTTGGTGTTCAG 3′; miR-484: 5′ TCA GGCTCAGTCCCCTCCCGAT 3′; miR-21-5p, 5′ TAGCTTATCAGACTGATGTTGA 3′, while the kit supplied oligo dT primer mRQ was used as a reverse primer for miRNA specific reactions, and both forward and reverse primers that were used for the amplification of U6 snRNA, as per the manufacturer’s recommendations. Each PCR reaction included 10 μL of 2 × SYBR mix, 2 μL of cDNA, while the concentration of miRNA specific forward and mRQ general reverse primers were kept at 10 μΜ and the reaction volume was 20 μL. The amplification reactions were carried out in a StepOnePlus™ Real-Time PCR System (ThermoFisher Scientific, Carlsbad, CA, USA) at the following cycling conditions: 50 °C for 2 min, 95 °C for 2 min, 95 °C for 15 s and 60 °C for 40 s, the last two steps were repeated 40 times. All amplification reactions that resulted in a C_T_ < 36 and a single peak in melt curve analysis was considered to be detected.

## 3. Results

### 3.1. sEV Isolation and Characterization of PE-Derived sEVs

The initial centrifugation of the freshly isolated PE in a clot activator vacutainer depleted the samples from the cells and especially red blood cells, which were trapped by the silica layer of the tube, preventing their lysis. The following centrifugation step precipitated large protein aggregates [33], while Proteinase K treatment and a subsequent centrifugation step resulted in substantial pellets consisting of digested non-EV protein contaminants. Larger vesicles such as shed microvesicles (sMV) and apoptotic bodies were subsequently pelleted at a higher centrifugation speed. Next, the cleared sample was incubated overnight with the PEG and sEVs were precipitated. Finally, an RNase A digestion was used to remove any non-sEV RNA that has been deprotected by the prior proteinase K digestion. The ExoProK process is shown in Figure 1. All sEV pellets isolated demonstrated similar size and gel-like morphology for the same amount of starting material. The proteinase K digestion step at 60 °C was critical for the removal of contaminating proteins (Figure 2). When the Proteinase K digestion was performed at 37 °C, the resulting pellet was significantly smaller compared to the one observed at 60 °C, indicating reduced removal of non-EV contaminants. These observations were confirmed by the detection of serum albumin by Western blot, a non-EV protein that is typically abundant in pleural fluids [34]. When the digestion step at 60 °C is implemented, albumin is substantially depleted compared to 37 °C, while considerable amounts of albumin are co-precipitated when the Proteinase K step is omitted. (Figure 3A, Lanes 1, 3 and 4).

To confirm that the isolated pellets were indeed sEVs, we visualized their morphology by TEM, measured their size distribution by NTA and detected the TSG101 sEV protein marker by Western blot. The TEM images revealed the presence of nanoparticles with spherical shape surrounded by a membrane (Figure 4A). Particles with smaller diameter and irregular shape were also observed; such particles likely represent ruptured vesicles (Figure 3B). The NTA revealed that the isolated vesicles had homogeneous size distribution with an average diameter of 172 nm (Figure 5A). Western blotting analysis demonstrated the presence of endosomal protein TSG101 (Figure 3B). EVs were also isolated with a commercially available column affinity kit [23], as confirmed with TSG101 detection and TEM imaging (Figure 3B and Figure 4B). Western blot results show that affinity column removed serum albumin less efficiently (Figure 3, Lanes 1 and 2), and the TEM images reveal that impurities are still present (Compare Figure 4A and Figure 4B). The vesicles isolated by the commercial column also did not display a homogenous size distribution, as shown by the multiple NTA peaks (Figure 5B).

### 3.2. Detection of sEV-Derived RNAs

To demonstrate that our method can be used as a basis to quantify sEV-enriched RNA biomarkers in PEs, we isolated sEVs from 20 pleural fluid samples and detected miRNAs that are known effectors of EV-mediated signal transduction [8]. Three miRNAs, namely miR-29a-5p, miR-484 and miR-21-5p, were selected; miR-21-5p is currently the most studied oncomiR in a variety of cancers [35], while miR-29a and miR-484 are proposed serum EV biomarkers for non-small cell lung cancer (NSCLC) prognosis [36,37]. The detected C_T_ values demonstrate that sEV RNA isolation was efficient and the amount of material recovered was ample for standard qPCR analysis (Figure 6). RNA recovered from sEV derived from 1.5 mL of PE was sufficient for approximately 40 qPCRs. At least three out of four sEV RNA targets tested were detected in all samples; all qPCR amplified products were specific, demonstrating a single melt peak, respectively. Consistent with the findings of previous studies that report the enrichment of sEV-derived miR-484 in NSCLC patient plasma [37], the amplification of miR-484 resulted in average C_T_ values less than 30 cycles in the NSCLC samples. Additionally, we detected U6 snRNA, a well-characterized non-coding RNA known to be enriched in EVs from various biofluids, such as serum and urine [38] (Figure 6).

## 4. Discussion

Since the molecular analysis of vesicle-enclosed cargo from biological fluids can yield clinically relevant information, much focus has been set on both the development of vesicle isolation techniques and the discovery of biomarkers for a variety of diseases. In this regard, serum, plasma and urine have been receiving inappropriately large attention from both the scientific community and the industry due to the minimally invasive nature of their sampling. PE sampling is substantially more invasive to that of serum, plasma and urine. However, PEs are clinical manifestations of diseases that have advanced to their later stages, and they represent a relevant biofluid for vesicle-based liquid biopsy analyses. Further, PEs are in close physiological proximity to the lung and in direct contact with the pleural walls. As such, PEs are expected to be enriched in biomarker-containing sEVs that originate from diseased cells in the chest and especially in the pleural cavity, thus increasing the sensitivity for the diagnosis and monitoring of the diseases related to these organs. Moreover, the accumulation of pleural fluid is caused by a variety of malignancies including those of the lung and the pleura. To this end, thoracocentesis followed by cytological examination represent an established medical guideline; malignant PEs are routinely diagnosed in clinical practice with near perfect specificity but average sensitivity (approximately 60%), predominantly through the detection of cancer cells present in the fluid. However, PEs also include biomarker-containing EVs, which are underutilized in standard clinical practice [3,4,39,40].

In this work, we describe ExoProK, an inexpensive, robust and reliable methodology that can be seamlessly integrated in the clinic to isolate PE-derived small EVs by Proteinase K treatment and PEG precipitation. Complying with the requirements of the International Society of Extracellular Vesicles (ISEV) for the characterization of extracellular vesicles [6,41]: (a) we visualized single isolated vesicles by TEM; (b) we demonstrate that the isolated vesicles have a homogenous size distribution by NTA; and (c) provide evidence that the isolated nanoparticles are enriched in TSG101 compared to untreated PE. TSG101 is an sEV marker enriched in EVs originating from the endosomal pathway (ESCRT) with well documented role in the biosynthesis of EVs [42]. The method is useful for the detection of PE-derived EV biomarkers and may ameliorate the low sensitivity of PE cytological analysis.

The treatment with Proteinase K to digest non-EV proteins is performed prior to the precipitation, while the subsequent centrifugation removes the bulk of protein contaminants in the form of large pellets and improves the purity of the EVs. SEVs are exceptionally stable in a wide range of temperatures, varying from −80 to 60 °C for 24 h [43]. Based on the thermal stability of sEV and the increased activity of Proteinase K within 50–60 °C [44,45], we were able to deplete albumin, an abundant non-EV associated protein present physiologically in this fluid [46]. The removal of protein contaminants was confirmed by the depletion of serum albumin from the final sEV preparation as shown by Western blot analysis (Figure 3). Following ExoProK, the NTA results demonstrate that the isolated vesicles are homogenous, sized approximately 172 nm, showing that the method is efficient in removing protein contaminants. 

In addition to albumin, HDL and LDL represent other non-EV contaminants present in PE that are particularly resistant to high temperature and proteolysis. However, HDL have very similar density with sEVs, and it is difficult to deplete even with established methods, such as size exclusion chromatography and density gradient ultracentrifugation [14,47,48,49]. Such particles have also been shown to co-precipitate along with EVs following precipitation-based methods such as the one described [14]. However, it is worth mentioning that sEV isolation procedures that offer superior purity compared to precipitation based ones, such as size exclusion chromatography and density gradient ultracentrifugation, are also ineffective at depleting such particles, and only a combination of both methods can be effective in depleting them [14]. It should be noted that, following digestion by Proteinase K, the detection of transmembrane proteins should be avoided as they are expected to be degraded. Such proteins as EV domains of integrin-like proteins play a fundamental role in the binding of the sEV with the recipient cell, and their destruction is expected to irreversibly hamper their biological function as signal transduction vehicles.

Since serum albumin is among the most abundant non-EV protein contaminants in pleural fluid and Proteinase K has very broad specificity [44], the enzyme was also effective to digest the remaining of protein contaminants including non-EV RNA binding proteins. The deprotected non-EV cell-free RNA was made susceptible to a final RNase A step to avoid misidentification as EV RNA content [41]. We also find that TSG101, an established sEV marker, is enriched in our preparations and also resisted proteinase K digestion for 1 h at 60 °C, demonstrating the preservation of the isolated vesicular cargo (Figure 3B).

A well-established EV isolation method is based on column affinity, which has also been used to isolate exosomes towards the detection of sEV derived RNA from PEs [23]. Further, ultracentrifugation-based methods, such as density gradient ultracentrifugation, are among the purest sEV isolation methods. However, the high cost equipment and its maintenance, the laboriousness of the method that requires specialized personnel and the limited number of samples for simultaneous analysis are factors that render ultracentrifugation impractical for routine use in clinical laboratories [14]. We also observed that ultrafiltration, an established alternative to ultracentrifugation, is unsuitable for sEV isolation, since the 100 kDa cut-off filters instantly and irreversibly clog when used to process pre-cleared pleural fluids. 

To demonstrate that our method can be used as a basis for quantifying enriched RNA biomarkers in PEs, we isolated sEV from 20 PEs and detected three miRNAs, miR-29a-5p, miR-484 and miR-21-5p, and a longer U6 snRNA by qPCR. miR-21-5p is among the best studied oncomiRs with established functions in a variety of cancers, including NSCLC [5,35]. Similarly, it has been reported to be enriched in plasma-derived sEV, known to mediate lung cancer metastasis and proposed as a prognosis screening biomarker [50,51,52]. miR-29a and miR-484 have also been proposed as prognosis biomarkers for NSCLC [36,37]. Additionally, miR-484 is enriched in plasma sEV in tuberculosis [53]. U6 snRNA, on the other hand, has been used as an internal reference gene for the relevant quantification of cellular miRNA due to its stable expression [38]. U6 snRNA has also been reported to be a stably enriched sEV derived from serum and urine, and thus it has been utilized as an internal reference gene for the quantification of sEV RNA cargo in these biofluids [38]. Our results show that at least three sEV-enriched miRNAs and U6 snRNA were successfully detected from 1.5 mL of NSCLC PE at average C_T_ values less than 30 in every sample tested, demonstrating the reliability of ExoProK.

Conclusively, in this study, we developed a method for the isolation of EVs from pleural fluid that is based on a preliminary Proteinase K digestion followed by PEG precipitation of sEV. The method complies with the requirements of the ISEV for the characterization of sEVs; it is affordable, scalable and easy to perform; it requires no specialized equipment; and it is efficient and pure enough for the detection of sEV-derived miRNAs, thus making it a robust choice for the detection of validated biomarkers in clinical routine.

## Figures and Tables

**Figure 1 mps-04-00031-f001:**
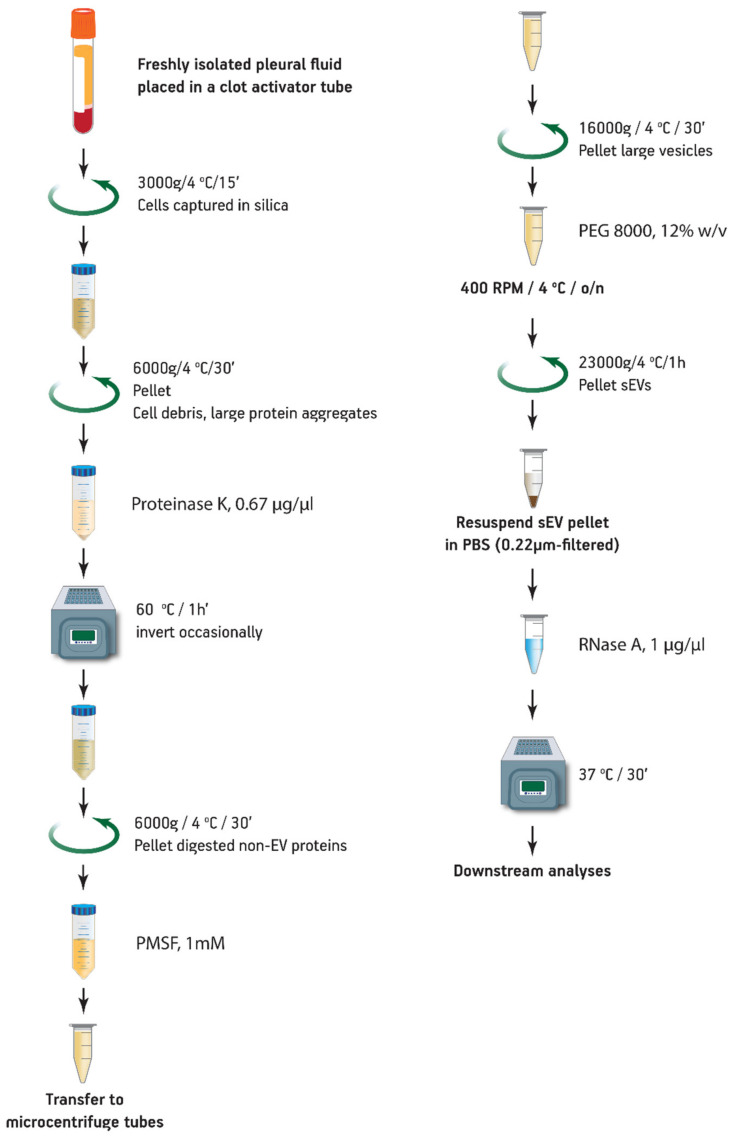
sEV isolation procedure. The scheme was designed with Adobe Illustrator^®^.

**Figure 2 mps-04-00031-f002:**
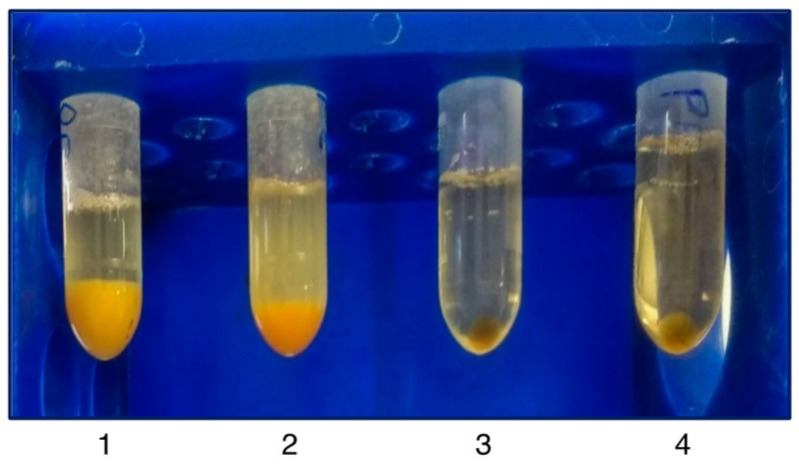
Morphology of pellets after Proteinase K digestion and precipitated sEVs. Two PE samples were subject to Proteinase K digestion and subsequent centrifugation (tubes 1 and 2). sEV pellets as visualized after the final centrifugation step (tubes 3 and 4) following the process described in Figure 1.

**Figure 3 mps-04-00031-f003:**
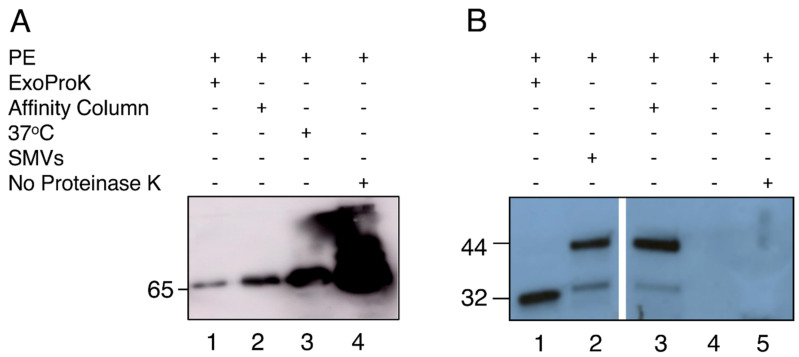
Characterization of isolated sEVs. (**A**) Depletion of albumin. Lane 1, sEV isolated with ExoProK; Lane 2, EVs isolated by affinity column [21]; Lane 3, sEVs isolated by ExoProK performed at 37 °C instead of 60 °C; Lane 4, PE processed with ExoProK without Proteinase K treatment. (**B**) Detection of TSG101 sEV marker of endosomal origin. Lane 1, sEV isolated with ExoProK; Lane 2, larger vesicles (200–500 nm) pelleted at 16,000 g at 4 °C/30 min; Lane 3, EVs isolated by affinity column [23]; Lane 4, cell-depleted pleural fluid; Lane 5, PE processed with ExoProK without Proteinase K treatment.

**Figure 4 mps-04-00031-f004:**
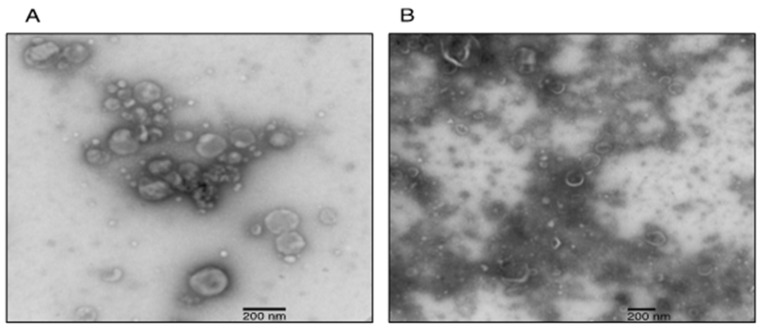
TEM characterization of EVs isolated by: ExoProK (**A**); and affinity column (**B**).

**Figure 5 mps-04-00031-f005:**
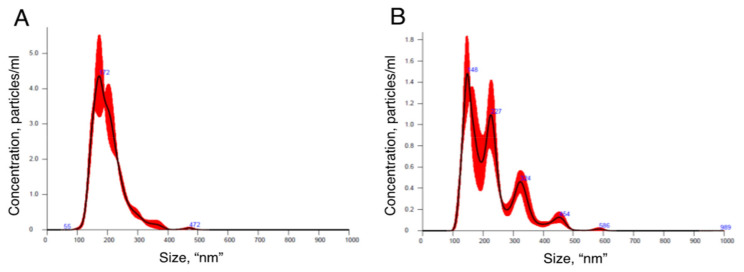
Size distribution of EVs isolated by: ExoProK (**A**); and affinity column (**B**).

**Figure 6 mps-04-00031-f006:**
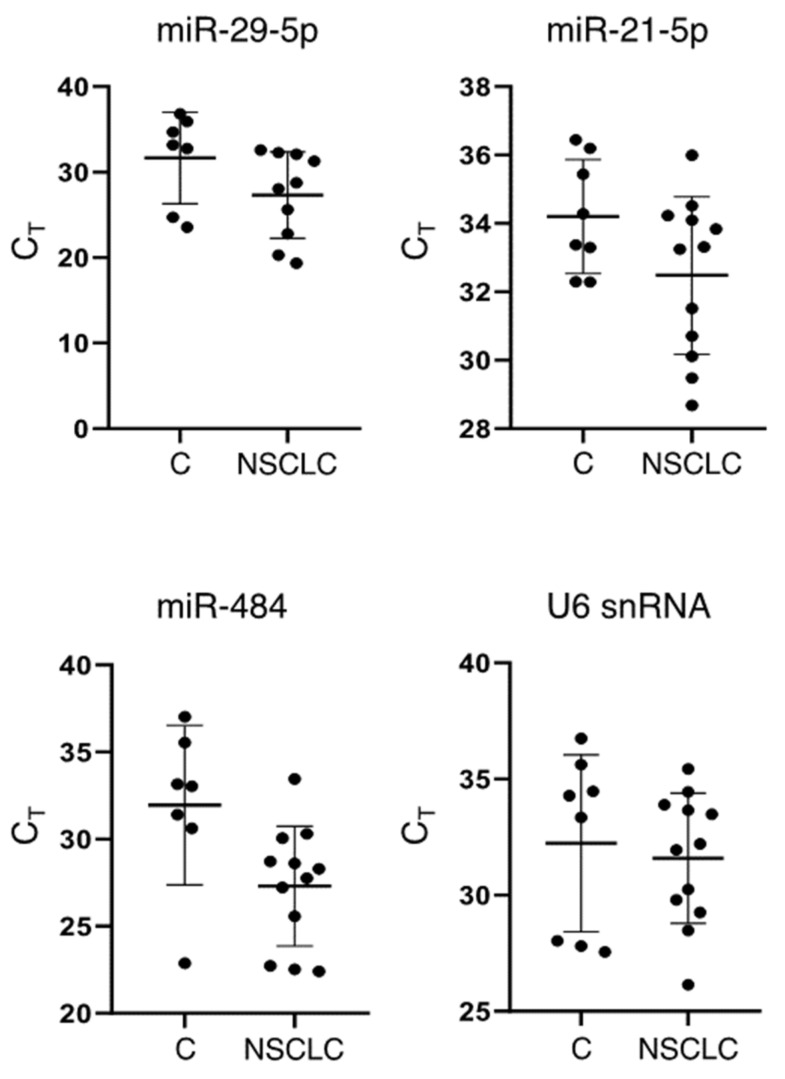
Detection of sEV derived RNAs with ExoProK. miR-21-5p, miR-29-5p, miR-484 and U6snRNA detected by qPCR in sEVs purified by PEs from NSCLC patients (NSCLC) and non-NSCLC subjects (C).

**Table 1 mps-04-00031-t001:** Published studies of pleural fluid-derived vesicles.

Vesicle Isolation Technique	Downstream Analysis	Project	Publication
Differential Ultracentrifugation	DNA—qPCR	Detect EGFR mutations in sEV enriched DNA	[20]
Sucrose gradient Ultracentrifugation	Protein Mass-spec	Protein biomarker discovery	[21]
Polymer precipitation	DNA—qPCR	Validate diagnostic value EGFR mutations	[22]
Affinity column	RNA—qPCR	RNA biomarker discovery	[23]
Sucrose gradient Ultracentrifugation	Protein Mass-spec	Protein biomarker discovery	[24]
Differential Ultracentrifugation	RNA sequencing	RNA biomarker discovery	[25]
Polymer precipitation	RNA—qPCR	RNA biomarker validation	[26]
Sucrose gradient Ultracentrifugation	Cell migration—proliferation	Determine the effect of PE sEV on cultured cells	[27]
Differential Ultracentrifugation	DNA sequencing	Detect common oncogene mutations in sEV-enriched DNA	[28]
Differential Ultracentrifugation	RNA sequencing	RNA biomarker discovery—validation	[29]

**Table 2 mps-04-00031-t002:** Clinicopathological characteristics of non-small cell lung cancer (NSCLC) patients and control subjects.

	NSCLC Patients *n = 12*	Control Subjects (non-NSCLC-Related) *n = 8*
Age, years (median)	72	65
Male/Female	8/4	5/3
Exudates	12	8
Smoking status		
Smoker/Ex-smoker	11	6
Non-smoker	1	2
Control Subjects’ Characteristics		
Tuberculosis	-	1
Post-operative	-	1
Kidney disease	-	1
Inflammation	-	2
Heart failure	-	1
Rheumatoid arthritis	-	1
Connective tissue disease	-	1

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
