# Peer review of "ExoProK: A Practical Method for the Isolation of Small Extracellular Vesicles from Pleural Effusions"

_mps, 2021, doi:10.3390/mps4020031_

Round 1

Reviewer 1 Report

Thank you for the effort and expertise with relevance to the revised manuscript and to maintain the high standards for peer-reveiwed journals.

Author Response

Thank you for your comments.

Reviewer 2 Report

Authors answered to all my concerns. However, I still have some major and minor issues that I listed below.

  • Scale bar in the figure 4 is missing.
  • Authors should use “exosome” term instead of small extracellular vesicles (sEVs) to avoid confusion in the reader.
  • Please at page two authors correct size of exosomes with <200 nm.
  • In the text, authors please correct PMM with MPM (Malignant Pleural Mesothelioma).
  • Please correct the cargo of exosomes. They contain lipids too.
  • The uptake of exosomes doesn’t not occur only via receptor-ligand, in fact there are other mechanisms involved as integrin adhesion, endocytosis as well as mechanisms still undefined. Please authors revised this concept.
  • Authors should explicit the NSCLC acronym in the text and in the table.
  • Please provide the total number of patients involved in the section 2.1. Additionally, in the text authors should clarify that patients with malignant disease are actually NSCLC affected and for the other subjects please enlist as example which kind of disease they have.
  • Please check the exact number of patients enclosed in the analysis. In the section 3.2 as well as in the discussion the pleural fluids samples are 22 but the total number of patients are 20.
  • In the caption of figure 5 please authors explicit that that is NTA analysis. Furthermore, please check the measurement unit in the graph: concentration for NTA analysis should be expressed as number of particles/mL. Please in the graph authors put in round brackets “nm” and check the scale bar used for the concentration.
  • Please authors explicit caption of Figure 6.

Authors should verify the presence of CD91, CD171, CD151, CD317 and TSPAN8 as potential exosomal biomarkers of NSCLC. They were found in bloody exosomes but it would be interesting if a correlation among the two fluids, blood and PE, were found.

Author Response

We would like to thank all the Reviewers for their efforts and the creative comments that contribute significantly to the overall improvement of our submitted manuscript. We took full account of all the points that were raised by Reviewer 2. Accordingly, we have extensively revised the text, redesigned one figure, and included 3 new references. All changes in the resubmitted
manuscript referring to Points of the Reviewers have been highlighted and labelled with comments in the margins to indicate the relevant Point. Please see attachment.

Round 2

Reviewer 2 Report

Authors replied to all of my concerns. Manuscript can be published in the present form.

This manuscript is a resubmission of an earlier submission. The following is a list of the peer review reports and author responses from that submission.

Round 1

Reviewer 1 Report

The study led by Antonopoulos et al., has attempted to come up with a simplified protocol that is easy and inexpensive to characterize EVs derived from pleural effusions by combining proteinase K and PEG precipitation methods. They have validated their protocol on the clinical samples by quantifying the miRNA cargo. It is a novel concept that would have an impact in the clinical set up. However, the study lacks some of the controls that can strengthen their effort and claim and they are listed below.

Major

  1. Although the authors have evaluated the role of proteinase K in the removal of albumin contamination that is abundant in pleural effusions, they have not shown if that influences the quality and recovery of the EV miRNA cargo. It would add more value to the manuscript to see the miRNA quantification with and without proteinase K treated EVs.

Minor

  1. Authors have mentioned why they chose 60 degrees Celsius for the protocol however they haven’t mentioned if they checked any other incubation time points for proteinase K treatment like 30 min or 2 hr to see if that provides better results and to justify more on the selection of 1 hr.
  2. Authors should mention the right concentration of Proteinase K used in this protocol.

In section 2.2. sEV isolation, authors mentioned proteinase K concentration to be 1mg/ml but in Fig.1 they mention it to be 1ng/ul instead 1microgram/ul.

  1. Authors can use the units for centrifugation uniformly rather than using both RCF and RPM at different sections of the manuscript.
  2. Text in Fig.1 are harder to read at this low resolution. Can be improved.

Reviewer 2 Report

The development of extracellular vesicles (EV) isolation methods are practical to screen routinely for cargo biomarkers in clinical laboratories. The authors indicate that these methods must be reliable, inexpensive, automated, capable of processing large amounts of samples and to be performed by non-specialized personnel. The techniques need to be inexpensive, efficient and straightforward methodologies to isolate EVs is by precipitation after the addition of volume excluding polymer solutions such as polyethylene glycol (PEG). The  authors report that the disadvantage of these isolation techniques is the low purity of the resulting EV pellets and co-precipitation of non-EV macromolecules and non-EV contaminants.  The authors describe a practical extracellular vesicle (EV) isolation method from pleural effusions (PEs) for the screening of EV RNA biomarkers that is based on a Proteinase K treatment step to digest the contaminants prior to standard polyethylene-glycol precipitation. The efficiency of the method is confirmed by transmission electron microscopy, nanoparticle tracking analysis and western blotting. The authors demonstrate the reliability and sensitivity of the method towards the detection of EV-enriched RNA biomarkers from multiple PEs. The method is based on two previous studies  and uses conditions to ensure maximum efficiency of EV isolation. The authors compared the effectiveness of their method (ExoProK) with a column-affinity based method for the discovery and detection of sEV RNA biomarkers (miR-29a-5p, miR-484, miR-21-5p, U6 snRNA).

Comments:

  1. A conclusion section may be important to this manuscript.
  2. The use of the methodology may be to assess the early onset of lung disease with relevance to the release of EVs composition (size/ structure) that are different from the middle and later stages of lung disease. The methodology may provide identification of EVs that are drug carriers versus EVs (miRNA). The method may be of clinical use to lung diseases with comparison between plasma and PEs extracellular vesicles with relevance to plasma HDL and early lung disease diagnosis. Plasma HDL is defective in many chronic diseases.
  3. The authors may comment on the effectiveness of this screening method and if it can compare EVs to high density lipoproteins (HDL). Interest with relevance to HDL (7.3-8nm) alterations in lung disease and lung cancer may also be relevant to lung disease progression and diagnosis. Extracellular vesicles and HDL exchange various components and may include proteins, metabolites, drugs, RNA species and nucleic acids. Does the method separate HDL from EVs and determine the role of HDL versus EVs with relevance to lung disease and cancer?

RELEVANT REFERENCES:

  1. Fessler MB.  Next Stop for HDL: the Lung. Clin Exp Allergy. 2012 Mar; 42(3): 340–342.
  2. Simonsen JB. What Are We Looking At? Extracellular Vesicles, Lipoproteins, or Both? Circ Res. 2017 Sep 29;121(8):920-922.
  3. Mohan, A., Agarwal, S., Clauss, M. et al. Extracellular vesicles: novel communicators in lung diseases. Respir Res 21, 175 (2020).

Reviewer 3 Report

In this manuscript, authors describe a method to isolate exosomes from pleural effusion (PE) using Proteinase K to remove protein contamination before to proceed with the classic PEG precipitation. They characterize exosomes with NTA analysis, Western blotting and TEM to confirm their origin. Even if the argument is certainly interesting and worthy of further studies because pleural effusion exosomes are poorly investigated in the literature, I recommend to reject the paper in this form. However, I listed below some suggestions to authors to improve the work.

  • To make the study statistically significant, authors should increase number of patients involved.
  • There is no novelty to use Proteinase K digestion to clean EVs from contaminants. At best, this work could be the first to use this method in PE matrix. Thus, work should be revised conceptually.
  • As authors said in the text, QPCR reference control is not reliable because extremely variable among samples. Authors should find another QPCR control in order to use RQ value instead of Ct.
  • Authors assert that method proposed fulfils criteria to integrate EV-based liquid biopsy as part of routine clinical practice. Nevertheless, operative procedure is too much time required. Maybe this method, if it will be optimized to have reliable results, could be used to find PE exosomal proteins and ncRNAs biomarkers for lung cancer in PE.
  • Conclusions are missing.